# Learning Neural Contracting Dynamics: Extended Linearization and Global Guarantees

**Sean Jaffe**[1,2*]**, Alexander Davydov**[1]**, Deniz Lapsekili**[2]**, Ambuj K. Singh**[2]**, and Francesco Bullo**[1]

[1] Center for Control, Dynamical Systems and Computation, University of California, Santa Barbara.
[2] Department of Computer Science, University of California, Santa Barbara.

## Abstract

Global stability and robustness guarantees in learned dynamical systems are essential to ensure well-behavedness of the systems in the face of uncertainty. We present Extended Linearized Contracting Dynamics (ELCD), the first neural network-based dynamical system with global contractivity guarantees in arbitrary metrics. The key feature of ELCD is a parametrization of the extended linearization of the nonlinear vector field. In its most basic form, ELCD is guaranteed to be (i) globally exponentially stable, (ii) equilibrium contracting, and (iii) globally contracting with respect to some metric. To allow for contraction with respect to more general metrics in the data space, we train diffeomorphisms between the data space and a latent space and enforce contractivity in the latent space, which ensures global contractivity in the data space. We demonstrate the performance of ELCD on the high dimensional LASA, multi-link pendulum, and Rosenbrock datasets.

## 1 Introduction

Due to their representation power, deep neural networks have become a popular candidate for modeling continuous-time dynamical systems of the form

$$\dot{x} = \frac{dx}{dt} = f(x), \tag{1}$$

where $f(x)$ is an unknown (autonomous) vector field governing a dynamical process. Beyond approximating the vector field $f$, it is desirable to ensure that the learned vector field is well-behaved. In many robotic tasks like grasping and navigation, a well-behaved system should always reach a fixed endpoint. Ideally, a learned system will still stably reach the desired endpoint even when pushed away from demonstration trajectories. Additionally, the learned system should robustly reach the desired endpoint in the face of uncertainty. In tasks such as manufacturing, animation, and human-robot interaction, functionality and safety require the learned system must smoothly follow a specific trajectory to its target.

To enforce stability guarantees, a popular approach has been to ensure that the learned dynamics admit a unique equilibrium point and have a Lyapunov function, e.g. [30]. While popular, approaches based on Lyapunov functions typically struggle to provide general robustness guarantees since global asymptotic stability does not ensure robustness guarantees in the presence of disturbances. Indeed, input-to-state stability of global asymptotically stable dynamical systems needs to be separately established, e.g., see Chapter 5 in [21].

To ensure robustness and to allow for smooth trajectory following, there has been increased interest in learning contracting dynamics from data [6]. A dynamical system is said to be contracting if any

---

*Corresponding author: `sjaffe@ucsb.edu`

38th Conference on Neural Information Processing Systems (NeurIPS 2024).

two trajectories converge to one another exponentially quickly with respect to some metric [27]. If a learned contracting system that admits a demonstration trajectory is pushed off that trajectory, it will follow a new trajectory that exponentially converges to the demonstration trajectory. Additionally, if a system is contracting, it is exponentially incrementally input-to-state stable [39]. If the system is autonomous, it also admits a unique equilibrium, is input-to-state stable, and has two Lyapunov functions that establish exponential stability of the equilibrium [9]. Since establishing contractivity globally requires satisfying a matrix partial differential inequality everywhere, prior works have focused on establishing contractivity in the neighborhood of training data [35, 36].

## 1.1 Related Works

**Stable, but not necessarily contracting dynamics.** Numerous works have aimed to learn stable dynamical systems from data including [7, 15, 22, 30, 32, 40, 45]. In [22], the authors introduce the Stable Estimator of Dynamical Systems (SEDS), which leverages a Gaussian mixture model to approximate the dynamics and enforce asymptotic stability via constraints on learnable parameters. In [30], the authors jointly learn the dynamics and a Lyapunov function for the system and project the dynamics onto the set of dynamics which enforce an exponentially decay of the Lyapunov function. This projection is done in closed-form and establishes global exponential convergence of the dynamics to the origin. In [40], the authors introduce Imitation Flow where trajectories are mapped to a latent space where states evolve in time according to a stable SDE. In [32], Euclideanizing Flows is introduced where the latent dynamics are enforced to follow natural gradient dynamics [2]. A similar approach is taken in [45] where additionally collision avoidance is considered.

**Existing works on learning contracting dynamics.** Learning contracting vector fields from demonstrations has attracted attention due to robustness guarantees [6, 33, 35, 37, 39]. In [33], the dynamics are defined using a Gaussian mixture model and the contraction metric is parametrized to be a symmetric matrix of polynomial functions. Convergence to an equilibrium is established using partial contraction theory [41]. In [35], the dynamics are defined via an optimization problem over a reproducing kernel Hilbert space; the dynamics are constrained to be locally contracting around the data points, i.e., there may be points in state space where the dynamics are not contracting. In [37] and [39], the authors study controlled dynamical systems of the form $\dot{x} = f(x, u)$, where $u$ is a control input and train neural networks to find a feedback controller such that the closed-loop dynamics are approximately contracting, i.e., contractivity is not enforced but instead lack of contractivity is penalized in the cost function.

Recent work, [6], has proposed a Neural Contractive Dynamical System (NCDS) which learns a dynamical system which is explicitly contracting to a trajectory. NCDS parametrizes contracting vector fields by learning symmetric Jacobians and performing line integrals to evaluate the underlying vector field. NCDS is computationally costly because of this integration. Additionally, the Jacobian parametrization used is overly restrictive, because not all contracting vector fields have symmetric Jacobians. Constraining the vector field to have symmetric Jacobian is equivalent to enforcing that the vector field is a negative gradient flow and contracting with respect to the identity metric [42]. For scalability to higher dimensions, NCDS leverages a latent space structure where an encoder maps the data space to a lower-dimensional space and enforces contracting dynamics in this latent space. Then a decoder "projects" the latent-space dynamics to the full data space. It is then argued that on the submanifold defined by the image of the decoder, the dynamics are contracting.

## 1.2 Contributions

In this paper, we present a novel model for learning deep dynamics with global contraction guarantees. We refer to this model as Extended Linearized Contracting Dynamics (ELCD). To the best of our knowledge, ELCD is the first model to ensure global contraction guarantees. To facilitate the development of this model, we provide a review of contracting dynamics and extended linearization of nonlinear dynamics. Leveraging extended linearization, we factorize our vector field as $f(x) = A(x, x^*)(x - x^*)$, where $x^*$ is the equilibrium of the dynamics. We enforce negative definiteness of the symmetric part of $A(x, x^*)$ everywhere and prove (i) global exponential stability of $x^*$, (ii) equilibrium contractivity of our dynamics to $x^*$, and (iii) using a converse contraction theorem, contractivity of the dynamics with respect to some metric.

Since negative definiteness of the symmetric part of $A$ is not sufficient to capture all contracting dynamics, we introduce a latent space of equal dimension as the data space and learn diffeomorphisms between the data space and this latent space. The diffeomorphisms provide additional flexibility in the contraction metric and allow learning of arbitrary contracting dynamics compared to those which are solely equilibrium contracting.

Our example in Section 3.4 provides theoretical justification for why the diffeomorphism and the learned contracting model must be trained jointly. In summary, if the diffeomorphism is trained first, and transforms the data to a latent space, the model class may not be expressive enough to accurately represent the true latent dynamics. If the diffeomorphism and dynamics are trained simultaneously, this limitation is overcome. This is in contrast to [6] which trains the diffeomorphism independently as a variational autoencoder and then trains the model after on the transformed data.

Our model, ELCD, directly improves on NCDS in several ways. We parameterize the vector field directly instead of parametrizing its Jacobian. Doing so prevents us from needing to integrate the Jacobian to calculate the vector field and thus speeds up training and inference. ELCD is also more expressive than NCDS because it can represent vector fields with asymmetric Jacobians. Additionally, ELCD is guaranteed to be contracting to an arbitrary equilibrium point, either selected or learned, at all training steps. NCDS, in contrast, must learn the equilibrium point over the course of training. Additionally, NCDS learns an encoder and decoder for a lower-dimensional latent space and thus can only be contracting on the submanifold defined by the image of the decoder. From initial conditions that are not on this submanifold, NCDS may not exhibit contracting behavior. In contrast, since the latent space of ELCD is of the same dimension as the data space, we train diffeomorphisms that ensure global contractivity in the data space. ELCD exhibits better performance than NCDS at reproducing trajectories from the LASA [26], n-link Pendulum, and Rosenbrock datasets. We additionally compare against the Euclideanizing Flow [32], and Stable Deep Dynamics [30] models.

## 2 Preliminaries

We consider the problem of learning a dynamical system $\dot{x} = f(x)$ using a neural network that ensures that the dynamics are contracting in some metric. Going forward, we denote by $Df(x) = \frac{\partial f}{\partial x}(x)$ the Jacobian of $f$ evaluated at $x$. To this end, we define what it means for a dynamical system to be contracting.

**Definition 1.** *A contracting dynamical system is one for which any two trajectories converge exponentially quickly. From [27], for a continuously differentiable map $f : \mathbb{R}^d \to \mathbb{R}^d$, the dynamical system $\dot{x} = f(x)$ is contracting with rate $c > 0$ if there exists a continuously-differentiable matrix-valued map $M : \mathbb{R}^d \to \mathbb{R}^{d \times d}$ and two constants $a_0, a_1 > 0$ such that for all $x \in \mathbb{R}^n$, $M(x) = M(x)^\top$ and $a_0 I_d \succeq M(x) \succeq a_1 I_d$ and additionally satisfies for all $x$*

$$M(x)Df(x) + Df(x)^\top M(x) + \dot{M}(x) \preceq -2cM(x). \tag{2}$$

The map $M$ is called the *contraction metric* and the notation $\dot{M}(x)$ is shorthand for the $n \times n$ matrix whose $(i, j)$ entry is $\dot{M}(x)_{ij} = \nabla M_{ij}(x)^\top f(x)$. A central result in contraction theory is that any dynamical system $\dot{x} = f(x)$ satisfying (2) has any two trajectories converging to one another exponentially quickly [27, 29, 39]. Specifically, there exists $L \geq 1$ such that any two trajectories $x_1, x_2$ of the dynamical system satisfy

$$\|x_1(t) - x_2(t)\|_2 \leq Le^{-ct}\|x_1(0) - x_2(0)\|. \tag{3}$$

In other words, contractivity establishes exponential *incremental* stability [4].

We note that any contracting dynamical system enjoys numerous useful properties including the existence of a unique exponentially stable equilibrium and exponential input-to-state stability when perturbed by a disturbance. We refer to [27] for more useful properties of contracting dynamical systems and [9] for a recent monograph on the subject.

The problem under consideration is as follows: given a set of demonstrations $\mathcal{D} = \{(x_i, \dot{x}_i)\}_{i=1}^n$ consisting of a set of $n$ state, $x_i \in \mathbb{R}^d$, and velocity, $\dot{x}_i \in \mathbb{R}^d$, pairs, we aim to learn a neural network $f(x_i) = \dot{x}_i$ that parametrizes a globally contracting dynamical system with equilibrium point $x^*$, such that $f(x^*) = 0$. In essence, this task requires learning both the vector field and the contraction metric, $M$ such that they jointly satisfy (2).

## 2.1 Contracting Linear Systems

Suppose we assumed that the dynamical system we aimed to learn was linear, i.e., $\dot{x} = Ax$ for some matrix $A \in \mathbb{R}^{d \times d}$ and we wanted to find a contraction metric $M$ which we postulate to be constant $M(x) := M$ for all $x$. The contraction condition (2) then reads

$$M(A + cI_n) + (A + cI_n)^\top M \preceq 0, \tag{4}$$

which implies that $A$ has all eigenvalues with $\mathrm{Re}(\lambda(A)) \leq -c$, see Theorem 8.2 in [18]. In other words, for linear systems, contractivity is equivalent to stability.

Although the condition (4) is convex in $M$ at fixed $A$, the task of learning both $(A, M)$ simultaneously from data is not (jointly) convex. Instead, different methods must be employed such as alternating minimization. A similar argument is made in [30] in the context of stability and here we show that the same is true of contractivity.

## 2.2 Contractivity and Exponential Stability for Nonlinear Systems

In the case of nonlinear systems, contractivity is not equivalent to asymptotic stability. Indeed, asymptotic stability requires finding a continuously differentiable Lyapunov function $V : \mathbb{R}^d \to \mathbb{R}_{\geq 0}$ which is positive everywhere except at the equilibrium, $x^*$, and satisfies the decay condition

$$\nabla V(x)^\top f(x) < 0, \quad \forall x \in \mathbb{R}^d \setminus \{x^*\}. \tag{5}$$

One advantage of learning asymptotically stable dynamics compared to contracting ones is that the function $V$ is scalar-valued, while the contraction metric $M$ is matrix-valued. A disadvantage of learning asymptotically stable dynamics compared to contracting ones is that we are required to know the location of the equilibrium point beforehand and no robustness property of the learned dynamics is automatically enforced. To this end, there are works in the literature that enforce an equilibrium point at the origin and learn the dynamics and/or the Lyapunov function under this assumption [30]. In the case of contractivity, existence and uniqueness of an equilibrium point is implied by the condition (2) and it can be directly parametrized to best suit the data.

## 3 Methods

### 3.1 Motivation

The motivation for our approach comes from the well-known mean-value theorem for vector-valued mappings, which we highlight here.

**Lemma 2** (Mean-value theorem (Proposition 2.4.7 in [1])). *Let $f : \mathbb{R}^d \to \mathbb{R}^d$ be continuously differentiable. Then for every $x, y \in \mathbb{R}^d$,*

$$f(x) - f(y) = \left( \int_0^1 Df(\tau x + (1 - \tau)y)d\tau \right)(x - y). \tag{6}$$

If $y = x^*$ satisfies $f(x^*) = 0$, then the continuously differentiable map $f$ admits the factorization

$$f(x) = A(x, x^*)(x - x^*), \quad \text{where} \tag{7}$$

$$A(x, x^*) = \int_0^1 Df(\tau x + (1 - \tau)x^*)d\tau. \tag{8}$$

This factorization is referred to as *extended linearization* in analogy to standard linearization, i.e., for $x$ close to $x^*$, $f(x) \approx Df(x^*)(x - x^*)$. We remark that when $d \geq 2$, extended linearization is not unique, and for a given $f$, there may exist several $A$ such that $f(x) = A(x, x^*)(x - x^*)$. In other words, (8) showcases one valid choice for $A$ such that this factorization holds. Indeed, this nonuniqueness has been leveraged in some prior works, e.g. Section 3.1.3 in [39], to yield less conservative contractivity conditions.

Since we know that a contracting vector field admits a unique equilibrium point, $x^*$, we restrict our attention to learning a matrix-valued mapping $A : \mathbb{R}^d \times \mathbb{R}^d \to \mathbb{R}^{d \times d}$ and ensuring that this mapping

has enough structure so that the overall vector field satisfies the contraction condition (2) for some contraction metric $M$. This task is nontrivial since $f(x) = A(x, x^*)(x - x^*)$ implies that

$$Df(x) = A(x, x^*) + \frac{\partial A}{\partial x}(x, x^*)(x - x^*), \tag{9}$$

where $\frac{\partial A}{\partial x} : \mathbb{R}^d \times \mathbb{R}^d \to \mathbb{R}^{d \times d \times d}$ is a third-order tensor-valued mapping. In what follows, we will show that a more simple condition will imply contractivity of the vector field. Namely, negative definiteness of the symmetric part of $A(x, x^*)$ will be sufficient for contractivity of the dynamical system $\dot{x} = A(x, x^*)(x - x^*)$.

## 3.2  Parametrization of $A(x, x^*)$

We show a simple example of our model, the Extended Linearized contracting Dynamics (ELCD). Let $x \in \mathbb{R}^d$ be the state variable and $f : \mathbb{R}^d \to \mathbb{R}^d$ be a vector field with equilibrium point $x^*$. As indicated, we parametrize our vector field by its extended linearization

$$\dot{x} = f(x) = A(x, x^*)(x - x^*), \tag{10}$$

where now

$$A(x, x^*) = -P_s(x, x^*)^\top P_s(x, x^*) \\ + P_a(x, x^*) - P_a(x, x^*)^\top - \alpha I_d \tag{11}$$

$P_s, P_a : \mathbb{R}^d \times \mathbb{R}^d \to \mathbb{R}^{d \times d}$ are neural networks, $I_d$ is the $d$-dimensional identity matrix, and $\alpha > 0$ is a constant scalar. Note that the symmetric part of $A(x, x^*)$ is negative definite since

$$\frac{A(x, x^*) + A(x, x^*)^\top}{2} = -P_s(x, x^*)^\top P_s(x, x^*) - \alpha I_d$$

and $P_s(x, x^*)^\top P_s(x, x^*)$ is guaranteed to be positive semidefinite.

We prove that a vector field parametrized this way is guaranteed to be (i) globally exponentially stable, (ii) equilibrium contracting as defined in [10], and (iii) contracting in some metric. The key tools are partial contraction theory [41] and a converse contraction theorem.

**Theorem 3** (Equilibrium Contraction and Global Exponential Stability)**.** *Suppose the dynamical system $\dot{x} = f(x)$ is parametrized as $f(x) = A(x, x^*)(x - x^*)$ where $A(x)$ is as in (11). Then any trajectory $x(t)$ of the dynamical system satisfies*

$$\|x(t) - x^*\|_2 \le e^{-\alpha t}\|x(0) - x^*\|_2. \tag{12}$$

*Proof.* We use the method of partial contraction as in [41]. Let $x(t)$ be a solution to $\dot{x}(t) = A(x(t), x^\star)(x(t) - x^*)$ with initial condition $x(0) = x_0$. Then, define the time-varying virtual system

$$\dot{y}(t) = A(x(t), x^*)(y(t) - x^*). \tag{13}$$

We will establish that this virtual system is contracting in the identity metric, i.e., (2) is satisfied with $M(x) = I_d$. We see that the Jacobian for this virtual system is simply $A(x(t), x^*)$ and

$$A(x(t), x^*) + A(x(t), x^*)^\top = -2P_s(x(t), x^*)^\top P_s(x(t), x^*) - 2\alpha I_d \\ \preceq -2\alpha I_d.$$

In other words, in view of (3), and since $M(x) = I_d$, any two solution trajectories $y_1(t)$ and $y_2(t)$ of the virtual system satisfy

$$\|y_1(t) - y_2(t)\|_2 \le e^{-\alpha t}\|y_1(0) - y_2(0)\|_2.$$

Note that we can pick one trajectory to be $y_1(t) = x^*$ for all $t$ and we can pick $y_2(t) = x(t)$. Since $x_0$ was arbitrary, this argument establishes the claim. □

Clearly, the bound (12) implies exponential convergence of trajectories of the dynamical system (10) to $x^*$. Moreover, this bound exactly characterizes equilibrium contractivity, as was defined in [10]. Notably, using the language of logarithmic norms, Theorem 33 in [10] establishes a similar result to Theorem 3 without invoking a virtual system.

Note that although Theorem 3 establishes global exponential stability and equilibrium contraction, it does not establish global contractivity. Indeed, the contractivity condition (2) is not guaranteed to hold with $M(x) = I_d$ without further assumptions on the structure of $A(x, x^*)$ in (10). Remarkably, however, due to a converse contraction theorem of Giesl, Hafstein, and Mehrabinezhad, it turns out that one can construct a state-dependent $M$ such that the dynamics are contracting. Specifically, since trajectories of the dynamical system satisfy the bound (12), for any matrix-valued mapping $C : \mathbb{R}^d \to \mathbb{R}^{d \times d}$ with $C(x) = C(x)^\top \succ 0$ for all $x$, the matrix PDE

$$M(x)Df(x) + Df(x)^\top M(x) + \dot{M}(x) = -C(x) \tag{14}$$

admits a unique solution for each $x$ [16]. In other words, the unique $M$ solving this matrix PDE serves as the contraction metric and will satisfy (2) with suitable choice for $c$. The following proposition provides the explicit solution for $M$ in terms of the solution to the matrix PDE.

**Proposition 4** (Theorem 2.8 in [16]). *Let* $C : \mathbb{R}^d \to \mathbb{R}^{d \times d}$ *be smooth and have* $C(x) = C(x)^\top$ *be a positive definite matrix at each* $x$. *Then* $M : \mathbb{R}^d \to \mathbb{R}^{d \times d}$ *given by the formula*

$$M(x) = \int_0^\infty \psi(\tau, x)^\top C(\phi(\tau, x))\psi(\tau, x)d\tau, \tag{15}$$

*is a contraction metric for the dynamical system* (10) *on any compact subset containing* $x^*$, *where* $\tau \mapsto \phi(\tau, x)$ *is the solution to the ODE with* $\phi(0, x) = x$ *and* $\tau \mapsto \psi(\tau, x)$ *is the matrix-valued solution to*

$$\dot{Y} = Df(\phi(t, x))Y, \quad Y(0) = I_d. \tag{16}$$

While it is challenging, in general, to compute the metric (15), numerical considerations for approximating it arbitrarily closely are presented in [17]. In practice, one would select $C(x) = I_d$ and evaluate all integrals numerically. For ELCD, unless one directly needs to know the contraction metric for application purposes, it is not required to compute the contraction metric at any point during either training or inference.

We remark that our parametrization for $A$ in (11) is similar to the parametrization for the Jacobian of $f$ that was presented in [6]. There are however a few key differences. Notably, $A(x, x^*)$ may be asymmetric while the Jacobian in equation (3) in [6] is always symmetric. Notably, since the Jacobian in [6] is symmetric and negative definite, the vector field $\dot{x} = f(x)$ is a negative gradient flow, $\dot{x} = -\nabla V(x)$, for some strongly convex function $V$. On the contrary, our dynamics (10) can exhibit richer behaviors than negative gradient flows in view of the asymmetry in $A(x, x^*)$. Additionally, since the dynamics in [6] with their parametrization can be represented as $\dot{x} = -\nabla V(x)$ for some strongly convex $V$, it is guaranteed to be contracting in the identity metric, $M(x) = I_d$ [42]. On the other hand, our dynamics (10) is not necessarily contracting in the identity metric and instead is contracting in a more complex metric given in (15).

### 3.3 Latent Space Representation

Realistic dynamical systems and their flows including the handwritten trajectories found in the LASA dataset, are often highly-nonlinear and may not be represented in the form (10) or obey the bound (12). One solution to these challenges is to transform the system to a latent, possible lower-dimensional, space and learn an ELCD in the latent space.

Latent space learning is possible because contraction is invariant under differential coordinate changes. From Theorem 2 of [29], given a dynamical system $\dot{x} = f(x)$, $f : \mathbb{R}^d \to \mathbb{R}^d$, with $f$ satisfying (1), the system will also be contracting under the coordinate change $z = \phi(x)$ if $\phi : \mathbb{R}^d \to \mathbb{R}^d$ is a smooth diffeomorphism. Specifically, if $\dot{x} = f(x)$ is contracting with metric $M$, then the system that evolves $z$ is contracting as well with metric given by $\tilde{M}(z) = D\phi(z)^{-\top} M(z)D\phi(z)^{-1}$, where $z = \phi(x)$ and $D\phi$ is the Jacobian of the coordinate transform. In other words, We can learn vector fields that are contracting in an arbitrary metric by training a vector field $f$ which is parametrized as (10) and using a coordinate transform $\phi$.

NCDS [6], treats the coordinate transform as a Variational Autoencoder (VAE) [23]. Their training procedure consists of two steps: first training the coordinate transform with VAE training, then training the function $f$ in the new learned coordinates.

Figure 1: The learned vector field and corresponding trajectories of an ELCD with no transform (left) and a model with a transform (right) when trained on data that is generated from a vector field that is contracting in a more general metric.

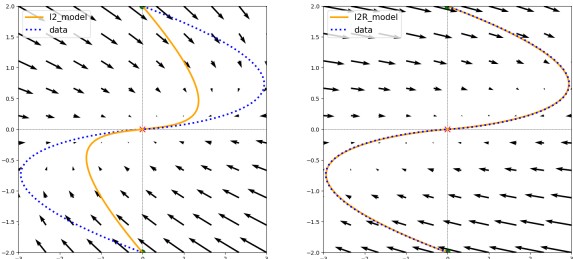

## 3.4 On the Interdependence of Diffeomorphism and Learned Dynamics

To demonstrate the need for a coordinate transform, we consider the task of learning a vector field that fits trajectories generated by the linear system $\dot{x} = Ax$ with $A = \begin{pmatrix} -1 & 4 \\ 0 & -1 \end{pmatrix}$. Clearly, this linear system cannot be represented in the form (10) with parametrization (11). To see this fact, we observe that $A + A^\top$ is not negative definite and thus no choice of $P_s$ or $P_a$ can represent this linear system. To remedy this issue, one can take the linear coordinate transform $z = \phi(x) = Px$, where $P = \begin{pmatrix} 1 & 0 \\ 0 & 4 \end{pmatrix}$. Then the $z$-dynamics read

$$\dot{z} = PAP^{-1}z = \begin{pmatrix} -1 & 1 \\ 0 & -1 \end{pmatrix} z \tag{17}$$

and now the symmetric part of $PAP^{-1}$ is negative definite and thus we can find suitable choices of $P_s, P_a$. Specifically, we can take $\alpha \in (0, 1/2)$, let $P_a(z) = \begin{pmatrix} 0 & 0 \\ 1/2 & 0 \end{pmatrix} z$, and let $P_s(z) = Qz$ where $Q$ is the matrix square root of $\begin{pmatrix} 1 & -1/2 \\ -1/2 & 1 \end{pmatrix} - \alpha I_2$. Note that in this toy example, asymmetry is essential to exactly represent these dynamics in the latent space, $z$. If we used the parametrization in [6], we would not be able to represent these dynamics since they have an asymmetric Jacobian. We demonstrate this routine in Figure (1). We generate two trajectories starting at $(0, 2)$ and $(0, -2)$, and use that data to train two ELCDs, one without and one with a learned, linear transform. Figure 1(a) shows the vector field and trajectories corresponding to an ELCD with no transform. The trajectories are forced to the center sooner than the actual data as a consequence of the bound (12). Learning a coordinate transform allows the ELCD to learn a system that is contracting in an arbitrary metric and exhibit overshoot. This is demonstrated by the learned system in Figure 1(b), which perfectly matches the data.

## 3.5 Choice of Diffeomorphism

There are several popular neural network diffeomorphisms including coupling layers [12, 32], normalizing flows [40], $\mathcal{M}$-flow [6, 8], spline flows [14], and radial basis functions [5].

A coupling layer $\phi : \mathbb{R}^d \to \mathbb{R}^d$ consists of a neural network $\theta : \mathbb{R}^k \to \mathbb{R}^N$ with $1 < k < d$ and a neural network $g : \mathbb{R}^N \times \mathbb{R} \to \mathbb{R}$. The transform $\phi$ maps input $x \in \mathbb{R}^d$ to $y \in \mathbb{R}^d$ with the following procedure:

(i) Set $y_i = x_i$ for $1 \leq i \leq k$ for some $1 < k < d$.
(ii) Set $y_i = g(x_{1:k}, x_i)$ for $k \leq i \leq d$

Coupling layers are invertible by doing the above process in reverse, so long as $g$ is invertible. A coupling layer can exhibit a wide variety of behaviors depending on the choice of $g$.

Polynomial spline curves are another diffeomorphism that are commonly used as $g$ in coupling layers. A polynomial spline has a restricted domain which is divided into bins. A different linear, quadratic

[31], or cubic [13] polynomial covers each bin. [14] introduce rational-quadratic splines, which constructs a spline from functions that are the quotient of two quadratic polynomials.

In practice, $\phi$ is the composition of several of transforms. Because the coupling transform only alters some coordinates, they are often used in conjunction with random permutations. If the latent space dimension is smaller than the data space dimension [6] makes the last composite function of $\phi$ an 'Unpad' function, which simply removes dimensions to match the latent space. The decoder $\phi$ is then prepended by a 'Pad' function, which concatenates the latent variable with the necessary number of 'zeros' to match the data space.

Of course, if the latent dimension is smaller than that of the data dimension, then $\phi$ can not be bijective. However, [6] argue that as long as $\phi$ is injective and has range over the dataset, then $f$ being contracting in the latent space implies that the learned dynamics are contracting on the submanifold defined by the image of $\phi^{-1}$. In other words, NCDS cannot be globally contracting. The consequences of this are shown in Figure (2). While NCDS can learn to admit an equilibrium point when on the submanifold, trajectories that fall off the submanifold may diverge. ELCD, in contrast, is contracting to the correct equilibrium point during all phases of training.

Figure 2: Plots of the vector fields and induced trajectories learned by ELCD (Top) and NCDS (Bottom) after different training epochs. ELCD is contracting always while NCDS may admit multiple equilibrium or diverge.

Epoch 1      Epoch 2      Epoch 3

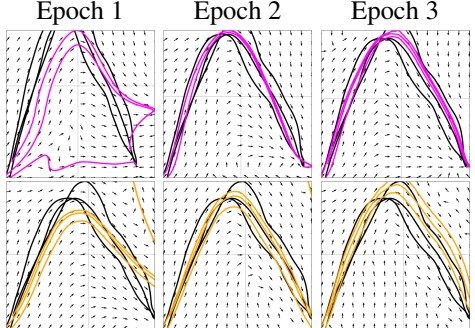

## 3.6 Training

Our task is to simultaneously learn the contracting system $f(x)$ and the metric in which $f$ contracts. As previously discussed, the metric is implicitly determined by $\phi(x)$. [6] uses a two-step method. First, they treat $\phi$ as a variational autoencoder and maximize the evidence lower bound (ELBO). They then fix the encoder and train the model to evolve the state in latent space. Note that the VAE objective is to make the encoded data to match a standard-normal distribution. Notably, VAE training does not encourage the encoder to transform the data to space in which the data corresponds to trajectories of a contracting system. If the data is not contracting in the transformed space, a contracting model will not be able to fully fit the data. This explains why, in our implementation of the two step-training, we are unable to reasonably learn the data. For further comparison with NCDS, we will train the encoder and model jointly.

# 4 Experiments

The code for our model and data can be found here: https://github.com/seanjaffe1/Extended-Linearized-Contracting-Dynamics. For all datasets we compare our method against NCDS [6], Stable Deep Dynamics (SDD) [30] and Euclideanizing Flow (EFlow) [32]. See A.2 for a detailed discussion of the methods. We report the dynamic time warping distance (DTWD) [20] (see A.1) and standard deviation between predicted and data trajectories. See A.5 for model implementation details. See also the Appendix for more details on these models.

Figure 3: Plots of the 2D LASA data. Demonstrations are in black. The learned ELCD trajectories in magenta are plotted along with the learned vector field. The vector field velocities have been normalized for visibility.

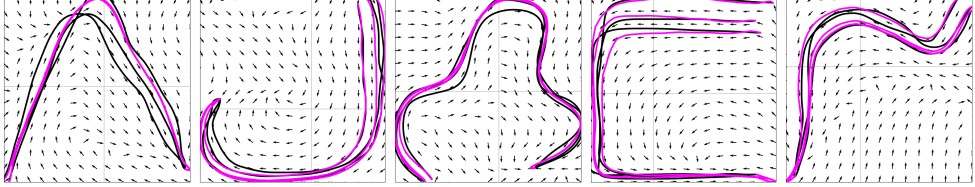

Table 1: Mean DTWD $\pm$ one standard deviation across 10 runs on LASA, multi-link pendulum, and Rosenbrock datasets

|  | SDD | EFlow | NCDS | ELCD |
| --- | --- | --- | --- | --- |
| LASA-2D | $0.37 \pm 0.32$ | $1.05 \pm 0.25$ | $0.59 \pm 0.61$ | $\mathbf{0.12 \pm 0.11}$ |
| LASA-4D | $2.49 \pm 2.4$ | $2.24 \pm 0.12$ | $2.19 \pm 1.23$ | $\mathbf{0.80 \pm 0.54}$ |
| LASA-8D | $5.26 \pm 0.50$ | $2.66 \pm 0.63$ | $5.04 \pm 0.77$ | $\mathbf{1.52 \pm 0.61}$ |
| Pendulum-4D | $0.49 \pm 0.11$ | $0.17 \pm 0.01$ | $1.35 \pm 2.26$ | $\mathbf{0.03 \pm 0.01}$ |
| Pendulum-8D | $0.75 \pm 0.08$ | $0.33 \pm 0.01$ | $2.88 \pm 0.69$ | $\mathbf{0.14 \pm 0.03}$ |
| Pendulum-16D | $1.86 \pm 0.14$ | $0.45 \pm 0.01$ | $1.65 \pm 0.31$ | $\mathbf{0.44 \pm 0.09}$ |
| Rosenbrock-8D | NaN | $1.90 \pm 0.16$ | $2.74 \pm 0.15$ | $\mathbf{1.22 \pm 0.01}$ |
| Rosenbrock-16D | NaN | $3.57 \pm 0.66$ | $3.68 \pm 0.12$ | $\mathbf{2.57 \pm 0.09}$ |

## 4.1 Datasets

We experiment with the LASA dataset [26], which consists of 30, two-dimensional curves. We use three demonstration trajectories of each curve. As in [6], the first few initial points of each trajectory are omitted so that only the target state has zero velocity. we stack two and four LASA trajectories together to make datasets of 4 and 8-dimensional trajectories, respectively. All data is standardized to have a mean of zero and variance of one. We use in total 10 2D curves, 6 4D curves, and 6 8D curves. Some 2D-LASA trajectories and their respective trained ELCD trajectories and vector fields are visualized in Figure (3).

We also experiment with simulated datasets. We simulate 6 trajectories of a 2,4, and 8-link pendulum (4D, 8D, and 16D respectively) each and 4 trajectories of 8D and 16D Riemannian gradient descent dynamics on a generalization of the Rosenbrock function (see Appendix A.4). Each model is trained on all trajectories of the same dimension, and then predictions are made starting from every initial point.

## 4.2 Results

Table 1 presents our results. ELCD performs the best in all tasks. These results shows the benefit of the increased expressiveness allowed by the skew symmetric component of our parametrization. These benefits are more apparent in the pendulum dataset. The oscillatory behavior of the pendulum dynamics is a product of complex eigenvalues in its Jacobian. Our model's skew symmetric component is what enables it to learn the pendulum dynamics so well. A sample of demonstration and learned pendulum trajectories is shown in figure (4).

The Rosenbrock dynamics are stiff and difficult to learn. In our training of SDD, we observed lack of stability and convergence, resulting in very large DTWDs. Hence, we report NaN for the results of the SDD models on the Rosenbrock dataset. While our model performs the best, there is still room for improvement. Handling such dynamics with multiple time scales is still an open challenge.

Figure 4: Phase plots of the two link-pendulum trajectories (blue) and the trajectories produced by ELCD (red). Each column is a different trajectory. The top row is the first link and the bottom row is the second link. The x-axis is angle and the y-axis is angular velocity.

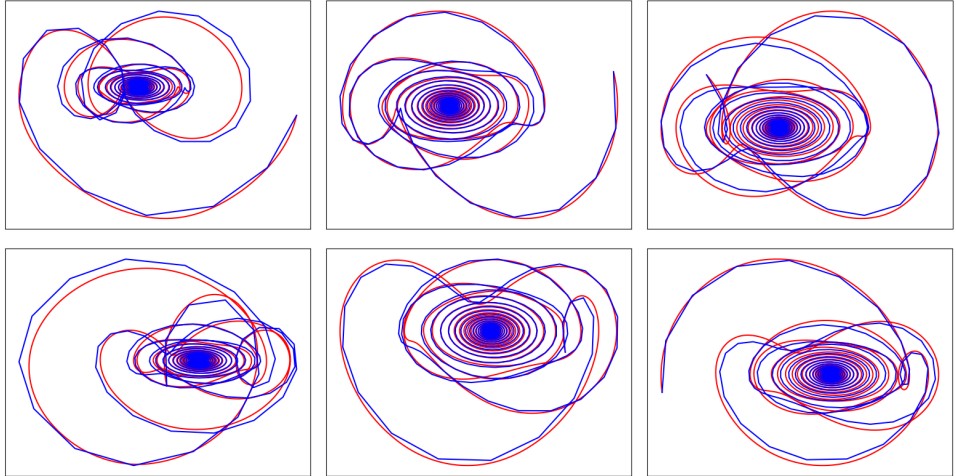

## 5   Limitations

Our method assumes that the underlying dynamics of the data trajectories are contracting. However, the diffeomorphism allows for a wide class of stable systems to be represented. Our method is currently implemented for trajectories that converge to the same fixed point. But, this can be overcome by manually changing the fixed point, depending on which trajectory the input data came from. Also, right now our method is limited to dynamics in $\mathbb{R}^n$, but we imagine that an extension to more general manifolds should be possible.

As we are motivated by applications in imitation learning and robotics, in this work, we are primarily interested in problems where the underlying systems should be robustly stable, especially away from training data. For this reason, we focus on learning dynamics which are guaranteed to be globally contracting and do not address learning other types of systems, such as those with multiple fixed points, limit cycles, or chaotic attractors. We imagine that extensions of ELCD could capture these richer dynamical behaviors by leveraging weaker notions of contraction including local contraction (i.e., contractivity in the region of attraction of a stable equilibrium), $k$-contraction (contraction of $k$-dimensional bodies) [43], transverse contraction [28] and contraction in the Hausdorff dimension [44]. Moreover, the notion of translation invariance mentioned could be studied using semicontraction theory [11], i.e., contraction to a subspace.

## 6   Conclusions

In this paper, we introduce ELCD, the first neural network-based dynamical system with global contractivity guarantees in arbitrary metrics. The main theoretical tools are extended linearization, equilibrium contraction, and a converse contraction theorem. To allow for contraction with respect to more general metrics, we use a latent space representation with dimension of the latent space equal to the dimension of the data space and train diffeomorphisms between these spaces. We highlight key advantages of ELCD compared to NCDS as introduced in [6], including global contraction guarantees, expressible parametrization, and efficient inference. We demonstrate the performance of ELCD on high-dimensional LASA datasets and simulated multi-link pendulum and Rosenbrock dynamics. Our method shows consistent performance across all datasets.

## Acknowledgments and Disclosure of Funding

This work was supported in part by the NSF Graduate Research Fellowship under grant 2139319, the AFOSR under award FA9550-22-1-0059, the NSF under grant no. 2229876, the Department

of Homeland Security, and by IBM. Any opinions, findings, and conclusions or recommendations expressed in this material are those of the author(s) and do not necessarily reflect the views of the NSF or its federal agency and industry partners.

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

# A  Appendix

## A.1  Metric

We evaluate trajectories by comparing them against their reference trajectory using the normalized dynamic time warping distance (DTWD) [20]. For two trajectories $x = \{x_i\}_{i \in \{1,...,t_1\}}$, $\bar{x} = \{\bar{x}_i\}_{i \in \{1,...,t_2\}}$ and some distance function $d(\cdot, \cdot)$, let DTWD be

$$\textbf{DTWD}(x, \bar{x}) = \frac{1}{t_1} \sum_{i=1}^{t_1} \Big( \min_{\bar{x}_j \in \bar{x}} d(x_i, \bar{x}_j) \Big) + \frac{1}{t_2} \sum_{i=1}^{t_2} \Big( \min_{x_j \in x} d(\bar{x}_i, x_j) \Big)$$

## A.2  Comparisons to Other Models

In this section, we describe the models that we compare to, namely EFlow [32], SDD [30], and NCDS [6].

**Euclideanizing Flow:** Starting with EFlow, they parametrize their dynamical system by

$$\dot{x} = -G_\psi(x)^{-1} \nabla \Phi(\psi(x)), \tag{18}$$

where $\psi : \mathbb{R}^d \to \mathbb{R}^d$ is a diffeomorphism, $\Phi : \mathbb{R}^d \to \mathbb{R}$ is a convex potential function, and $G_\psi(x) = D\psi(x)^\top D\psi(x)$ is the induced Riemmanian metric. The dynamics (18) then has the interpretation of being the steepest descent for $\Phi \circ \psi$ on the Riemmanian manifold defined by metric $G_\psi$. Specifically, in [32], the convex potential function is defined to be $\Phi(y) = \|y - y^\star\|_2$, where $y^\star = \psi(x^\star)$ and the diffeomorphism is parametrized via a coupling layer [12]. Note that contraction properties of dynamics of these form were studied in [42]. Since $\Psi$ is chosen to be convex but not strongly convex, these dynamics are only *weakly contracting*, i.e., satisfy (2) with $c = 0$.

**Stable Deep Dynamics:** In [30], the authors parametrize both an unconstrained vector field $\hat{f} : \mathbb{R}^d \to \mathbb{R}^d$ and a Lyapunov function $V : \mathbb{R}^d \to \mathbb{R}_{\geq 0}$ via neural networks. Then to enforce global exponential stability, at every point $x \in \mathbb{R}^d$, they project $\hat{f}(x)$ onto the convex set of points $\{u \in \mathbb{R}^d \mid \nabla V(x)^\top u \leq -\alpha V(x)\}$. They do this projection in closed-form so that the dynamics have the representation

$$\dot{x} = \hat{f}(x) - \nabla V(x) \frac{\text{ReLU}(\nabla V(x)^\top \hat{f}(x) + \alpha V(x))}{\|\nabla V(x)\|_2^2}. \tag{19}$$

Under a smart parametrization of $V$ using input-convex neural networks [3], the authors guarantee global exponential stability of (19) to the origin with rate $\alpha$. Note that the dynamics (19) are continuous, but not necessarily differentiable. Due to this potential nonsmoothness, it is theoretically unknown whether these dynamics are contracting (since Theorem 2.8 in [16] requires at least some differentiability).

**Neural Contractive Dynamical Systems:** In [6], the authors enforce contractivity by directly enforcing negative definiteness of the Jacobian matrix of the vector field by parametrizing

$$Df(x) = -(J_\theta(x)^\top J_\theta(x) + \epsilon I_d), \tag{20}$$

where $J_\theta : \mathbb{R}^d \to \mathbb{R}^{d \times d}$ is a neural network. Under this parametrization of the Jacobian, it is straightforward to see that (2) holds with $c = \epsilon$ and $M(x) = I_d$ for all $x$. To recover the vector field from the Jacobian, the fundamental theorem of calculus for line integrals (which can be seen as a version of the mean-value theorem) is utilized to express

$$\dot{x} = f(x) = f(x_0) + \int_0^1 Df((1-t)x_0 + tx)(x - x_0)dt, \tag{21}$$

where $x_0$ and $f(x_0)$ denote the initial condition and its velocity, respectively.

To express richer dynamics, NCDS also consists of an encoder and decoder for a lower-dimensional latent space. The dynamics in the latent space are constrained to be (21) and the encoder and decoder are parametrized using a VAE. Since, generally, the latent space is of lower dimension than the data space, the resulting dynamics are only guaranteed to be contracting on the submanifold defined by the image of the decoder.

### A.3 Universal Representation

Our ELCD model equipped with a coupling layer-based transformation can represent any contracting dynamical system. It is known that any smooth nonlinear system with a hyperbolically stable fixed point can be exactly linearized inside its basin of attraction via a suitable diffeomorphism [25]. Effectively, our extended linearization parameterization is tasked with learning the stable linear part, while the coupling layers aim to learn and approximate this suitable diffeomorphism. As our ELCD parametrization can universally approximate any linear model, we simply need our coupling layer to universally approximate all diffeomorphisms. Indeed, [38] proved that the coupling layers are universal approximators for diffeomorphisms.

### A.4 Additional Details on Datasets

In this section, we provide additional elaboration on the multi-link pendulum dataset and the Rosenbrock dataset.

**Multi-link pendulum:** The multi-link pendulum is a simple mechanical system which is the interconnection of $n$ rigid links under the force of gravity. We specifically assume that there is damping on each of these links, which makes the resulting dynamical system stable and almost all trajectories converge to the downright equilibrium point. In this case, the state of the pendulum is described by the $n$ pairs $(\theta_i, \dot{\theta}_i)$, where $\theta_i$ is the angle of the $i$-th link and $\dot{\theta}_i$ is its angular velocity. Thus, this dynamical system is $2n$ dimensional.

As was done in [30], we adapt the code from http://jakevdp.github.io/blog/2017/03/08/triple-pendulum-chaos/ to generate data for $n$ links.

**Rosenbrock datset:** The classical Rosenbrock function, see [34]

$$f(x, y) = (1 - x)^2 + 100(y - x^2)^2, \tag{22}$$

is a nonconvex function with global minimum at $(1, 1)$. Despite its apparent nonconvexity, it is known that a Riemannian gradient descent dynamics, is contracting with respect to a suitable Riemannian metric, see Example 3 in [42] for details. In optimization and in machine learning, the Rosenbrock function is a benchmark for the design of various optimizers and in the study of Riemannian optimization [19, 24].

For a higher-dimensional generalization of the Rosenbrock function, we consider

$$f(x) = \lambda_1 (1 - x_1)^2 + \sum_{i=2}^{n} \lambda_i (x_i - x_{i-1}^2)^2, \tag{23}$$

where $\lambda_i > 0$ are parameters that affect the conditioning of the function. This generalization is still nonconvex and admits several saddle points. Note that the classic Rosenbrock corresponds to $n = 2, \lambda_1 = 1, \lambda_2 = 100$. The unique global minimum of this generalization is at $(1, 1, \ldots, 1)$. The corresponding contracting Riemmanian gradient descent dynamics that find this global minimum are

$$\dot{x} = -G(x)^{-1} \nabla f(x), \tag{24}$$

where $G(x) = D\psi(x)^\top D\psi(x)$, where $\psi : \mathbb{R}^n \to \mathbb{R}^n$ is the mapping $\psi(x) = (\sqrt{\lambda_1}(1 - x_1), \sqrt{\lambda_2}(x_2 - x_1^2), \ldots, \sqrt{\lambda_n}(x_n - x_{n-1}^2))$. Note that these dynamics were designed via a generalization of the procedure proposed in [42]. To generate data, we choose four initial points and evolved them according to (24).

### A.5 Model Details

For the encoder $\phi$, we use the composition of two coupling layers composed of rational-quadratic spline. The splines cover the range $\{-10, 10\}$ with 10 bins, and linearly extrapolated outside that range. The parameters of each spline are determined by residual networks, each containing two transform blocks with a hidden dimension of 30. A permutation and linear flow layer is placed before, in the middle, and after the two quadratic spline flow layers. For ELCD, we let the latent dimension size equal the data dimension size. $P_a$ and $P_s$ are implemented as two-layer neural networks with a hidden dimension of 16.

As the authors of NCDS have not yet made their code publicly available, we implemented NCDS to the best of our abilities. We used a latent dimension of size 2 for all datasets. This is in-line with the description in [6], and necessary given the extra cost from integrating the Jacobian in higher-dimensional latent spaces. We use the same encoder $\phi$ as for ELCD with an added un-padding layer as the last function which removes the last $d - 2$ dimensions.

## A.6 Training details

All experiments are trained with a batch size of 100, for 100 epochs, with an Adam optimizer and learning rate of $10^{-3}$. All computation is done on CPUs.

NCDS [6] trained the model and the encoder to output $x_{t+1}$ from $x_t$ using a single Euler integration step: $x_{t+1} = \phi^{-1}(\phi(x_t) + dt * f(\phi(x_t)))$. Their loss was the MSE between the predicted and true $x_{t+1}$. We empirically find better performance by training our model to directly predict the velocity vector $\dot{x}_{t+1} = \phi_J^{-1}(x_t)f(\phi(x_t))$, where $\phi_J^{-1}(x_t)$ is the Jacobian inverse of $\phi(x_t)$. Calculating the gradient of $\phi_J^{-1}(x_t)$ efficiently required a slight alteration to the original $\mathcal{M}$-flow [8] code, which we have included in our repository.

