# OpenReview forum: "Learning Neural Contracting Dynamics: Extended Linearization and Global Guarantees"
_NeurIPS.cc/2024/Conference — NeurIPS 2024 poster_

### Official Review · Reviewer_Njwt · 2024-07-06

**Soundness:** 4
**Presentation:** 3
**Contribution:** 3
**Rating:** 8
**Confidence:** 4

**Summary:**

The paper proposes a novel neural network architecture, Extended Linearized Contracting Dynamics (ELCD), for learning dynamical systems with global contractivity guarantees. The authors leverage the concept of extended linearization to factorize the vector field and enforce negative definiteness of its symmetric part, ensuring global exponential stability and equilibrium contraction. To extend contractivity to more general metrics, they introduce a latent space representation and learn diffeomorphisms between the data and latent spaces. The method is evaluated on various datasets, demonstrating superior performance compared to existing models, including Neural Contractive Dynamical Systems (NCDS).

**Strengths:**

The paper addresses the important problem of learning stable and robust dynamical systems, which has significant implications in various fields.

The proposed ELCD model is theoretically sound, with guarantees of global contractivity, a property that ensures robustness to perturbations.

The use of extended linearization and latent space representation is innovative and allows the model to capture a wider range of contracting dynamics.

The experimental results are promising, showing that ELCD outperforms existing methods on several benchmark datasets.

ELCD addresses some limitations of NCDS. Specifically, ELCD parameterizes the vector field directly, avoiding the computationally expensive Jacobian integration step in NCDS. ELCD also allows for asymmetric Jacobians, leading to increased model expressivity compared to NCDS, which is restricted to symmetric Jacobians.

ELCD trains the encoder jointly with the dynamics model, allowing the encoder to learn representations specifically tailored for capturing contracting dynamics. This is in contrast to NCDS, which trains the encoder separately as part of a Variational Autoencoder (VAE) before training the dynamics model. The joint training in ELCD potentially leads to better performance by optimizing the encoder for the specific task of representing contracting dynamics.

**Weaknesses:**

The assumption of contractivity in the underlying dynamics might limit the applicability of the model to real-world systems where this assumption may not hold.

The current implementation is restricted to trajectories converging to a fixed point and dynamics in Euclidean space, which could be extended to more general settings. This limitation is particularly evident in the LASA handwriting dataset, where the model struggles to capture the translation invariance inherent in human handwriting. In reality, the end-point location of handwriting is not fixed, and the entire letter can be translated without affecting its validity.  Also, handwriting can often exhibit overlapping trajectories, which cannot be modeled by the current implementation. Moreover, the learned model can only generate one handwriting.

It is unclear why the joint training of the encoder and the model is possible for ELCD but not in NCDS. The authors could address this point to clarify the advantages of their approach.

The paper lacks a clear discussion of the specific types of problems where ELCD would be most applicable and where it might not be suitable. A more detailed analysis of the model's strengths and weaknesses in different scenarios would be beneficial.

**Questions:**

1. Is the ELCD model expressive enough to represent all contracting dynamical systems? If not, what are the limitations, and can they be formally characterized?

2. The paper states that "contraction is invariant under differential coordinate changes..." (L227) but also mentions that "If the data is not contracting in the transformed space..." (L291). These statements seem contradictory. Can the authors clarify this discrepancy and explain how the choice of diffeomorphism affects the contractivity of the learned dynamics?

3. The computational complexity of approximating the contraction metric is mentioned as a potential limitation. Can the authors elaborate on the scalability of the method to high-dimensional systems and suggest possible strategies to mitigate this issue?

4. The paper focuses on trajectories converging to a fixed point. How can the method be extended to handle more complex dynamical behaviors, such as limit cycles or chaotic attractors?

5. Why is joint training of the encoder and the model possible for ELCD but not in NCDS?

**Limitations:**

The limitation sections address some of the limitations of the current approach, but is insufficient.

The current implementation of ELCD is limited to trajectories converging to a fixed point and dynamics in Euclidean space. This restricts its applicability to problems where these assumptions hold. For example, while ELCD is applied to model handwriting in the LASA dataset, it struggles to capture the translation invariance inherent in human handwriting, where the end-point location is not fixed. It would be beneficial to  discuss the specific types of problems where ELCD would be most applicable and where it might not be suitable.

---

> ### Author Rebuttal · Authors · 2024-08-07
>
> **It is unclear why the joint training of the encoder and the model is possible for ELCD but not in NCDS.**
>
> It is possible to jointly train the encoder and the model for NCDS as well. The authors of NCDS treat the encoder as a Variational Autoencoder (VAE) and train it by minimizing the evidence lower bound. The authors claim that the VAE training helps deal with the complexity of high dimensional systems. We argue, however, that the encoder and model must be trained jointly. If the encoder is trained independently of the model, then it is not necessarily true that the dynamics will be $l_2$-contracting in the learned transformed space. We confirmed this lack of contractivity experimentally, finding that VAE training of the encoder followed by training of the model did not produce reportable results. These poor results explain why we used joint training for NCDS for our baseline results. Since the NCDS code is not available, we have reproduced it to the best of our abilities.
>
>
> **The paper lacks a clear discussion of the specific types of problems where ELCD would be most applicable...**
>
> We envision applications in imitation learning where the resulting learned
> model has global guarantees of contraction, even away from the training
> data. Additionally, we imagine learning control systems of the form $\dot{x} = f(x) +
> Bu$, where $f$ is contracting. We could then leverage recent results on
> optimal control of contracting systems to design effective controllers for
> these systems [7].  We will update the introduction,
> motivation, and discussion about future research to highlight these points.
>
>
> **Is the ELCD model expressive enough to represent all contracting dynamical systems?**
>
> Thank you for your question.
>
> It is known that any smooth nonlinear system with a hyperbolically stable
> fixed point can be exactly linearized inside its basin of attraction via a
> suitable diffeomorphism [8].
>
> In essence, our extended-linear parameterization is tasked with learning
> the stable linear part, while the coupling layers aim to learn and
> approximate this suitable diffeomorphism.
>
> So your question can be transcribed into the following: are coupling layers
> capable of approximating any diffeomorphisms?
>
> Some results in this direction were obtained
> by [9,10]. The survey [11] confirms that those universality conditions hold for our choice in coupling layer [12].
>
> However, the universality results in [9,10] apply to learning transformations between cumulative distribution functions. At this time, we cannot directly answer the universality question about diffeomorphisms.
>
>
> **Can the authors clarify... how the choice of diffeomorphism affects the contractivity of the learned dynamics?**
>
> You raise a good point. If we have a contracting dynamical systems, then any diffeomorphism applied to the system ensures that the resulting systems remains contracting (in a different Riemannian metric, however). In L291 we were commenting on the specific parametrization of NCDS [1]. Since the NCDS parametrization ensures contraction in the $l_2$ metric, it is possible that an arbitrary diffeomorphism applied to data coming from some contracting dynamics is not contracting in the $l_2$ metric (although it would still be contracting in some other metric). We realize this point was not fully clear in the first version of our paper and
> the revision will include a clearer more detailed explanation.
>
>
>
> **Complexity of approximating the contraction metric...**
>
> This is a good question. In our opinion, ELCD scales well independently of the complexity of approximating the contraction metric. This is because we do not compute the contraction metric at any step during training or evaluation. The complexity of approximating the metric would only arise if the end user needed to have an expression for it; something we do not need in our tasks.

---

> > ### Comment · Reviewer_Njwt · 2024-08-13
> >
> > I thank the authors for the clarification in their response.  I'm keeping the original score.

---

### Official Review · Reviewer_UYYA · 2024-07-13

**Soundness:** 3
**Presentation:** 3
**Contribution:** 3
**Rating:** 6
**Confidence:** 2

**Summary:**

The paper presents Extended Linearized Contracting Dynamics (ELCD), a dynamical system with neural network components that has global contraction guarantees. They demonstrate improved efficiency and performance on the trajectory fitting LASA dataset (upto 8 dim), pendulum dataset (upto 16 dim), Rosenbrock dataset (upto 16 dim).

**Strengths:**

1. Strong theoretical basis for the ELCD model.

2. Well-written and easy-to-read paper.

3. Reasonably sized experiments for a paper with a primarily theoretical contribution.

**Weaknesses:**

1. If the authors would like to present more of a empirical contribution, it would help to scale the experiments to something of the order of 100s of dimensions such as in common RL environments like mujoco-humanoid.

**Questions:**

Please see above.

---

> ### Author Rebuttal · Authors · 2024-08-07
>
> First, we would like to thank you for finding our work to have a strong theoretical basis, well-written,
> and easy-to-read with reasonable experiments. Regarding larger-scale experiments, we have been unable to
> perform them at this time. We agree that they would add value and we envision executing them as future
> work.

---

> > ### Comment · Reviewer_UYYA · 2024-08-12
> >
> > Thank you. I will keep my score.

---

### Official Review · Reviewer_HPgs · 2024-07-23

**Soundness:** 3
**Presentation:** 3
**Contribution:** 3
**Rating:** 4
**Confidence:** 2

**Summary:**

The paper presents a method for learning contracting representations from a dynamical system. The novelty of the method lies in learning a linear map and a coordinate transform map (Diffeomorphism), which extends the types of contracting systems that can be learned. The experiments show that the proposed methods performs better empirically than Neural Contractive Dynamical System and other approaches for learning contracting dynamics.

**Strengths:**

- The paper is well structured and easy to follow
- The problem motivation beyond the vast amount of methods on learning stable dynamics is well formulated
- The relation to recent literature on this topic is also well done

**Weaknesses:**

The main weakness of the paper is the experimental part.
There are details missing and it feels "rushed", e.g., table caption unclear (see below).
Additionally, he set of tasks are very limited and don't go beyond the "usual suspects" in stable dynamical system learning (e.g. pendulum chains).

Besides the experiments, a point of limitation would be that the approach itself is by no means "groundbreaking" but rather incremental in the context of learning contracting dynamics (whether empirically it is truly groundbreaking cannot be assessed from the provided experiments)

**Questions:**

The caption in Table 1 misses which metric is represented here and also what the +- means (this should not be implicitly assumed)

**Limitations:**

The paper discusses some extensions to having multiple fixed points.
However, the paper lacks a true discussion on limitations and scenarios where the approach may fail (not necessarily due to the approach but also including due to learning it)

---

> ### Author Rebuttal · Authors · 2024-08-07
>
> **The novelty of the method lies in learning a linear map...**
>
> Thank you for your time in reviewing our paper. Respectfully, we would like to clarify that we do not learn a linear map, but rather a nonlinear matrix-valued map
> $x \mapsto A(x,x^*)$ via the parametrization (11) to allow for nonlinear
> contracting dynamics in the latent space.
>
>
>
> **The main weakness of the paper is the experimental part...**
>
> We do focus on the ``usual suspects" in stable dynamics learning since they
> are common benchmarks studied in other works. These examples allow us to
> showcase that ELCD outperforms other state-of-the-art methods for learning
> stable dynamics in established benchmarks or well-known systems.
> Beyond LASA and the multi-link pendulum, we argue
> that the Rosenbrock dataset is novel and valuable (while the Rosenbrock
> dataset has not been featured in other works on learning dynamics, the
> Rosenbrock function is a classic function in optimization and ML). The Rosenbrock dynamics are novel in that they are stiff and evolve at different timescales. In each case we are able to showcase the superiority of ELCD compared to prior alternative methods.
>
>
>
>
> **The approach itself is by no means ``groundbreaking" but rather incremental...**
>
> We would argue that the approach is more than incremental in nature as we
> allow for learning a larger class of contracting dynamics compared to NCDS
> by our novel parametrization which captures asymmetric Jacobian
> matrices. The benefits of our parameterization are clearest in our updated results on the multi-link pendulum using continuous-time training. The pendulum has inherently oscillatory dynamics, whose Jacobian has complex eigenvalues. No learning model with symmetric Jacobian can represent this system, regardless of what diffeomorphism is used. Our parameterization is key towards learning this system. The commonality of the multi-link pendulum (and more geneally oscillatory dynamics) in the dynamical systems literature demonstrates the value of our contribution. Moreover, by directly parametrizing the extended linearization,
> we avoid the expensive line integral necessary in NCDS.
> Finally, we feel that the recent improvement from discretized-time to continous-time training is also valuable.
> As a testament to
> benefits of these changes, we can see the notably improved performance of ELCD
> compared to existing benchmarks in our numerical experiments.
>
>
>
> **The caption in Table 1 misses which metric is represented here...**
>
> You are correct that we were not precise enough in our caption of Table 1. To
> clarify, we are reporting the dynamic time warping distance (DTWD) between
> the trajectories of the learned model and the trajectories of the ground
> truth dataset. The $\pm$ corresponds to one standard deviation from the
> mean. We will clarify these details before the final submission.

---

### Official Review · Reviewer_2hrV · 2024-07-25

**Soundness:** 4
**Presentation:** 4
**Contribution:** 3
**Rating:** 7
**Confidence:** 3

**Summary:**

This paper proposes a novel parameterization of the extended linearisation form of a dynamical system that guarantees global contractivity. Whilst the most basic form of this parameterization only ensures contractivity in some (implicitly defined) metric, a latent space version is also proposed that enables more flexible learning of dynamics that are contractive in more general metrics. Experiments are provided that show superiority over related methods.

**Strengths:**

This paper is very well written; it provides a very good intro / recap of contractivity which was very useful to me at least, and a clear articulation of what was missing from previous works on the subject (e.g. NCDS is computationally expensive and has overly constrained (symmetric) Jacobians, no global contraction guarantees, etc). The idea of directly parameterising the extended linearization of the system is very neat, and so is the use of the converse contraction theorem of Giesl et al to prove the main contractivity result. Although the experiments are a bit thin, I find them convincing enough.

**Weaknesses:**

The authors very quickly gloss over the motivations for learning contractive dynamics; I had to go read the NCDS paper to understand the real-world relevance of such dynamics. Let me write down below what occurred to me when reading the intro (which mostly betray my lack of experience in potential practical applications of this paper), and the authors may or may not want to modify their introduction to preempt this potential confusion.

My initial thoughts when reading the first paragraph of the intro, in particular “it is desirable to ensure [...]”:
- For one, one might actually be learning from data that comes from a dynamical system with multiple stable fixed points, so why is it desirable to enforce a single one in the learned model?
- Relatedly, why does the stability of the _learned_ dynamical system matter? I would assume that what typically matters in robotics/control is that the actual physical system implemented (and in particular, the closed control loop) be stable and robust? What does that have to do with the learned model? Is model stability important because an unstable model (where the ground truth is stable) would extrapolate very badly outside the region where there is data?

(I was thinking of a situation where deep learning is used to learn an internal model of a plant from observational data, e.g. for subsequent control purposes; in which case (i) autonomous dynamics are not necessarily very relevant and (ii) it's unclear why those dynamics would be contractive in general; now I understand that there are applications in e.g. imitation learning where it might be useful to ensure that the learned dynamics have such guarantees, but it was hard to guess from the current intro; the intro of the NCDS paper helped).

**Questions:**

In the partial contraction analysis of Eq 13, it might be worth stating that x can be any function of time (as opposed to a constant). This seems critical to the eventual conclusion, where you say "we can pick y2(t) = x(t)", as this statement implicitly relies on a trajectory of the original system also being a valid trajectory of the virtual system. I'm guessing this is very standard but as someone not familiar with this type of partial contraction analysis I thought I'd mention it.

A few typos
- l.41 “enforce an exponentially decay the Lyapunov function”
- l.52 “are constrainted”
- l.53 “there are points [...]” → there may be points [...]
- l.161 “prior words” → prior works
- l.258: diffeopmorphisms
- l.287 minimize the ELBO → maximize the ELBO
- l.290 the data is corresponds
- l.456: metrc
- just a suggestion for your acronym: "ELCoD" would be easier to pronounce ("ell-cod") than "ELCD"

**Limitations:**

Could the authors please comment on whether requiring contraction behaviour (and in particular, the existence of a unique stable equilibrium) might be too constraining in some applications?

I wonder if the authors could also say a few words about the controlled case -- i.e. driven systems as opposed to autonomous systems. E.g. if the autonomous system $\dot{x} = f(x)$ obeys the guarantees provided by this work, what guarantees might be automatically inherited by the corresponding control affine system $\dot{x} = f(x) + B u$?

---

> ### Author Rebuttal · Authors · 2024-08-07
>
> **Introduction**
>
> Thank you for clarifying your thought process. We will make the following changes to the introduction to make things clearer:
>
> We will replace line 17 onward in the first paragraph with:
>
> "Beyond approximating the vector field f, it is desirable to ensure that the learned vector field is well-behaved. In many robotic tasks like grasping and navigation, a well-behaved system should always reach a fixed endpoint. Ideally, a learned system will still stably reach the desired endpoint even when pushed away from demonstration trajectories. Additionally, the learned system should robustly reach the desired endpoint in the face of uncertainty. In other tasks, such as manufacturing, animation, and human-robot interaction, the learned system must smoothly follow a specific trajectory to its target. Following a trajectory is necessary to ensure function, as is the case with a manufacturing robot dynamically interacting with its environment, and safety, as is necessary with a robot occupying the same space as humans."
>
> We will additionally replace lines 27-28 in paragraph three with:
>
> "To ensure robustness and to allow for smooth trajectory following, there has been increased interest in learning contracting dynamics. A dynamical system is said to be contracting if any two trajectories converge to one another exponentially quickly with respect to some metric [25]. If a learned contracting system that admits a demonstration trajectory is pushed off that trajectory, it will follow a new trajectory that exponentially converges to the demonstration trajectory.
> "
>
>
>
>
>
> To address those specific concerns:
>
> **One might actually be learning from data that comes from a dynamical system with multiple stable fixed points**
>
> Thank you for bringing this point up. Indeed, contracting dynamics can only
> have one stable fixed point. In this work, we are assuming that the
> underlying dynamics are globally contracting (hence with a unique fixed
> point) as this assumption ensures strong convergence and robustness
> guarantees for the learned dynamics, even away from training data.
>
> As a possible extension to systems with multiple fixed points, one could
> imaging learning multiple ELCD models, one for the neighborhood of each
> fixed point and then (i) either devising a strategy to blend them for
> global problems, or (ii) depending on the initial condition, leveraging the
> specific relevant model.
>
>
>
> **Why does the stability of the learned dynamical system matter?**
>
> Like you mention, we envision applications in imitation learning where the
> resulting learned model has global guarantees of contraction, even away
> from the training data. We will update the introduction to be more specific
> as to why we want to learn globally contracting dynamics (and mention
> extensions to multiple equilibria and multiple local ELCD models).
>
>
>
> **In the partial contraction analysis...**
>
> Of course you are completely correct. We will be more precise and state
> that $x(t)$ is meant to be a specific solution trajectory of the original
> system $\dot{x}(t) = A(x(t),x^*)(x(t) - x^*)$.
>
>
>
> **Words about the controlled case...**
>
> Thank you for your curiosity on this topic. Our method can be naturally
> extended to learning control systems $\dot{x} = f(x) + Bu$ which are
> contracting in $x$ at fixed $u$. These systems automatically inherit an
> input-to-state stability property and an entrainment property, i.e., a
> periodic input $u$ results in a stable periodic solution $x$,
> see [6]. Moreover, control systems with a contracting
> drift term can be amenable to computationally efficient tools for optimal
> control, see [7]. We are actively studying the extension of
> ELCD to closed-loop contractivity guarantees, i.e., learning both $f$ and a
> closed-loop controller $u$ that offers global contraction guarantees (not
> just at the training data as is frequently studied in existing works).

---

### Author Rebuttal · Authors · 2024-08-07

**Explanation of Updated Results**

After submission, we improved the performance of our method by implementing a change in the training loss. We had previously trained the model $f$ and the encoder $\phi$ to output $x_{t+1}$ from $x_{t}$ using a single Euler integration step: $x_{t+1} = \phi^{-1}(\phi(x_t) + dt * f(\phi(x_t))$. Our loss was the MSE between the predicted and true $x_{t+1}$. This training method, which we call discretized-time training, is the same method used by [1] to train their NCDS model.

We have now updated our training method to, instead, directly predict the velocity vector $\dot{x}_{t+1} = \phi^{-1}_J(x_t)f(\phi(x_t)$, where $\phi^{-1}_J(x_t)$ is the Jacobian of the inverse of $\phi(x_t)$. We call this new training method continuous-time training. To avoid slowdowns, the computation of this Jacobian requires a careful implementation in pytorch; this will be documented in the final code. This change is purely technical and does not impact the parameterization of the model, the parametrization of the diffeomorphism, or the overall content of the paper. Directly predicting the velocity vector through continuous-time training allows us to better demonstrate the efficacy of our method. The new results are shown in table (1) in the attached document, which is the same as table (1) in the submission, with an additional column for the ELCD trained via continuous-time training. In the submission, we will remove the column from table (1) with ELCD trained by the original discretized-time training method. Our ELCD model with the new, continuous-time training loss has overall improved performance. Additionally, we visualize the performance of ELCD with continuous-time training on the 2-link pendulum in figure (1) in the attachment.

**Regarding learning non-contracting dynamics and limitations**

Several reviewers asked us to comment on limitations and learning tasks
where the true dynamics have multiple fixed points or limit cycles /
chaotic attractors. Motivated by applications in imitation learning and
robotics, in this work, we are primarily interested in problems where the
underlying systems should be robustly stable, especially away from training
data, following the same reasoning in [1]. For
this reason, we focus on learning dynamics which are guaranteed to be
globally contracting. We imagine that extensions of ELCD could capture
these richer dynamical behaviors by leveraging weaker notions of
contraction including local contraction (i.e., contractivity in the region
of attraction of a stable equilibrium), $k$-contraction (contraction of
$k$-dimensional bodies) [2], transverse
contraction [3] and contraction in the Hausdorff
dimension [4]. Moreover, the notion of translation
invariance mentioned by Reviewer Njwt could be studied using
semicontraction theory [5], i.e., contraction to a
subspace. Since we do not study these extensions here, we will add more
details in the limitations section elaborating on this point.

**References**

[1] H. Beik-Mohammadi, S. Hauberg, G. Arvanitidis, N. Figueroa, G. Neumann, and L. Rozo, “Neural
contractive dynamical systems,” in International Conference on Learning Representations, 2024.

[2] C. Wu, I. Kanevskiy, and M. Margaliot, “k-contraction: Theory and applications,” Automatica, vol. 136,
p. 110048, 2022.

[3] I. R. Manchester and J.-J. E. Slotine, “Transverse contraction criteria for existence, stability, and
robustness of a limit cycle,” Systems & Control Letters, vol. 63, pp. 32–38, 2014.

[4] C. Wu, R. Pines, M. Margaliot, and J.-J. E. Slotine, “Generalization of the multiplicative and additive
compounds of square matrices and contraction in the Hausdorff dimension,” IEEE Transactions on
Automatic Control, 2022.

[5] G. De Pasquale, K. D. Smith, F. Bullo, and M. E. Valcher, “Dual seminorms, ergodic coefficients, and
semicontraction theory,” IEEE Transactions on Automatic Control, vol. 69, no. 5, 2024. To appear.
[6] H. Tsukamoto, S.-J. Chung, and J.-J. E. Slotine, “Contraction theory for nonlinear stability analysis
and learning-based control: A tutorial overview,” Annual Reviews in Control, vol. 52, pp. 135–169,
2021.

[7] S. Taoufik and B. Missaoui, “Method of successive approximations for stochastic optimal control: Con-
tractivity and convergence,” arXiv preprint 2405.07048, 2024.

[8] Y. Lan and I. Mezi ́c, “Linearization in the large of nonlinear systems and Koopman operator spectrum,”
Physica D: Nonlinear Phenomena, vol. 242, no. 1, pp. 42–53, 2013.

[9] C.-W. Huang, D. Krueger, A. Lacoste, and A. Courville, “Neural autoregressive flows,” in International
Conference on Machine Learning, pp. 2078–2087, PMLR, 2018.

[10] P. Jaini, K. A. Selby, and Y. Yu, “Sum-of-squares polynomial flow,” in International Conference on
Machine Learning, pp. 3009–3018, PMLR, 2019.

[11] I. Kobyzev, S. J. D. Prince, and M. A. Brubaker, “Normalizing flows: An introduction and review of
current methods,” IEEE Transactions on Pattern Analysis and Machine Intelligence, vol. 43, no. 11,
pp. 3964–3979, 2021.

[12] C. Durkan, A. Bekasov, I. Murray, and G. Papamakarios, “Neural spline flows,” in Advances in Neural
Information Processing Systems (H. Wallach, H. Larochelle, A. Beygelzimer, F. d'Alch ́e-Buc, E. Fox,
and R. Garnett, eds.), vol. 32, Curran Associates, Inc., 2019

---

### Decision · Program_Chairs · 2024-09-25

**Decision:**

Accept (poster)

**Comment:**

### Summary

This paper proposes a method for learning a contractive dynamical system from demonstrations, with global contractivity guarantees. The system is flexible and parameterized by a simple neural network architecture that relies only on mild constraints. Contractivity is achieved under an implicit metric. Additionally, a diffeomorphism enables the modeling of the dynamics, as the system is learned in the latent space, where the representations help to ensure contractivity, while the mapping to the input space handles the expressivity.

$$$$

### Strengths

**Contribution**: This paper considers an important problem and addresses some limitations of the seminal NCDS model (efficiency, expressiveness, and global contraction guarantees). It introduces an innovative combination of latent space learning and extended linearization, enabling the capture of contracting dynamics through learned representations. The provided results are promising.

**Theoretical Clarity**: The paper is well-written with a clear structure (easy to follow). The theoretical guarantees and insights were appreciated by most reviewers. The concept of extended linearization is neat as well as the use of the converse contraction theorem to prove the main contractivity result.

$$$$

### Concerns

**Experiments**: Most reviewers found the experimental section to be somewhat limited, noting that it does not sufficiently demonstrate the claimed advantages of the proposed model.

**Novelty**: Concerns were raised by Reviewer HPgs, who described the proposed methodology as "incremental," noting that the benefits cannot be adequately assessed from the provided experiments.

$$$$

### Conclusion
Considering the contributions of the paper --particularly the modeling choices and theoretical results-- along with the efforts made to address the reviewers' questions and the additional experiments/extensions provided in the rebuttal, I believe this paper is a reasonable contribution to the conference and recommend acceptance as a poster. While the experimental section may be lacking, the other aspects were well-received by most reviewers, even though the approach appears somewhat incremental to previous works.